# Cone-Beam CT-Guided Lung Biopsies: Results in 94 Patients

**DOI:** 10.3390/diagnostics10121068

**Published:** 2020-12-10

**Authors:** Daniel Gulias-Soidan, Nilfa Milena Crus-Sanchez, Daniel Fraga-Manteiga, Juan Ignacio Cao-González, Vanesa Balboa-Barreiro, Cristina González-Martín

**Affiliations:** 1Department of Interventional Radiology, Complejo Hospitalario Universitario de A Coruña (CHUAC), As Xubias 84, 15006 A Coruña, Spain; daniel.gulias.soidan@sergas.es (D.G.-S.); nilfa.milena.crus.sanchez@sergas.es (N.M.C.-S.); daniel.fraga.manteiga@sergas.es (D.F.-M.); juan.cao.gonzalez@sergas.es (J.I.C.-G.); 2Clinical Epidemiology and Biostatistics Unit, Biomedical Research Institute of A Coruña (INIBIC), Complejo Hospitalario Universitario de A Coruña (CHUAC), SERGAS, University of A Coruña (UDC), As Xubias 84, 15006 A Coruña, Spain; vanesa.balboa.barreiro@sergas.es; 3Rheumatology and Public Health Research Group, Nursing Research and Health Care, Biomedical Research Institute of A Coruña (INIBIC), Complejo Hospitalario Universitario de A Coruña (CHUAC), SERGAS, University of Coruña (UDC), As Xubias 84, 15006 A Coruña, Spain

**Keywords:** lung, transthoracic biopsy, radiation

## Abstract

Background: The aim of this study was to evaluate the diagnostic capacity of Cone-Beam computed tomography (CT)-guided transthoracic percutaneous biopsies on lung lesions in our setting and to detect risk factors for possible complications. Methods: Retrospective study of 98 biopsies in 94 patients, performed between May 2017 and January 2020. To obtain them, a 17G coaxial puncture system and a Siemens Artis Zee Floor vc21 archwire were used. Descriptive data of the patients, their position at the time of puncture, location and size of the lesions, number of cylinders extracted, and complications were recorded. Additionally, the fluoroscopy time used in each case, the doses/area and the estimated total doses received by the patients were recorded. Results: Technical success was 96.8%. A total of 87 (92.5%) malignant lesions and 3 (3.1%) benign lesions were diagnosed. The sensitivity was 91.5% and the specificity was 100%. We registered three technical failures and three false negatives initially. Complications included 38 (38.8%) pneumothorax and 2 (2%) hemoptysis cases. Fluoroscopy time used in each case was 4.99 min and the product of the dose area is 11,722.4 microGy/m^2^. Conclusion: The transthoracic biopsy performed with Cone-Beam CT is accurate and safe in expert hands for the diagnosis of lung lesions. Complications are rare and the radiation dose used was not excessive.

## 1. Introduction

Percutaneous transthoracic biopsy (PTB) is a well-established and frequently used procedure for the diagnosis of lung nodules, regardless of their size [1,2,3]. Although PTB can be performed using fluoroscopic or ultrasound guidance, techniques guided by computed tomography (CT) and CT fluoroscopy have become generalized for the diagnosis of small lung lesions [2,3].

Both are precise and safe techniques [1,2,3]. Gerghty et al. [2] reported an accuracy of CT-guided PTB of 93.5% and Hiraki et al. [3] obtained an overall precision for CT-guided fluoroscopy of 95.2%. However, there are problems for performing biopsies with CT, such as the small diameter of the gantry, the limitation of guiding it in the puncture plane, or the exposure to radiation by the operators [4,5,6].

Relatively recently, Cone-Beam CT (CBCT) systems consisting of a C-arc, X-ray tube, and flat-panel-type detector have been introduced to the field of interventional radiology [7,8,9,10,11]. These systems offer great flexibility to orient the detector around the patient, provide CT images that can be reconstructed three-dimensionally, and allow real-time fluoroscopy. In addition, they facilitate access to the puncture point through an open arch and provide orthogonal coordinates to direct the biopsy needle. It is expected that all these advantages of CBCT improved the precision and efficacy of PTB, and different studies [6,11,12,13,14,15,16,17,18,19] have reported diagnostic accuracies between 91.7% and 98.2% in this regard. There are limited data in the literature on the radiation dose for CBCT-guided lung biopsy but the high potential radiation exposure of the operators during CT procedures is well-known [11]. The limitation of references in the literature where the relationship between radiation doses is described, comparing CT with fluoroscopy, motivated the provision of data on this topic.

The objective of this work was to evaluate the diagnostic capacity of the Cone-Beam CT-guided transthoracic percutaneous biopsy in lung lesions and to detect risk factors for possible complications. 

## 2. Materials and Methods 

From May 2017 to January 2020, we performed 98 CBCT-guided PTBs in 94 patients (mean age: 68.9 years; standard deviation: 9.1; age range: 42.4–89.2 years) using an 17G coaxial needle and automated biopsy system (Biopince. Argon Medical Devices, Dallas, TX, USA). Of the 94 patients, 4 had to undergo a repeat biopsy: in 3 of the cases the first sample was negative or insufficient (classified as false negatives (FNs)) and in 1 there was a problem in its processing. In our study, repeated biopsies were considered different initial cases of PTB.

There were 69 male and 25 female patients. The mean size of the 98 punctured lesions was 35.6 mm (range: 10–120 mm).

PTBs were performed using a CBCT system (DynaCT-Siemens Artis Zee Floor vc21. Siemens Medical Solutions, Erlangen, Germany) by three radiologists experienced in interventional procedures. After acquisition of the CT images, they were transferred to the workstation (Leonardo, Siemens Medical Solutions, Erlangen, Germany) to determine the safest and most effective puncture route, taking into account the precision and possible associated complications with the PTB. The virtual guide (iGuide, Siemens Medical Solutions) was used, performing an automatic vertical alignment of the arch from the point of entry into the skin to the target lesion (“bull’s-eye” projection). This system provides virtual marks on the fluoroscopic image to direct the puncture needle to the target marked on the source images (Figure 1 and Figure 2). After inserting a 17G coaxial introducer (Argon Medical Devices, Dallas, TX, USA), a control CBCT was performed to verify the location of the needle. If the tip of the coaxial needle was at the desired point, we performed the biopsy with an 18G needle (Biopince. Argon Medical Devices, Dallas, TX, USA) (Figure 3 and Figure 4).

Between 1 and 4 tissue samples were obtained depending on their appearance, the presence of pneumothorax and the patient’s ability to remain immobile. We then removed the coaxial introducer in an apnea and performed a new CBCT to identify complications related to the procedure (Figure 5, Figure 6 and Figure 7). The patients remained under surveillance for 30–45 min with constant control in the area adjacent to the examination room that we allocated for it.

After PTB, operators recorded descriptive patient data (age and sex), target characteristics (size, location and distance from the pleura to the target lesion) and procedure information (patient position, number of tissue samples, pneumothorax, hemoptysis, bleeding and radiation dose).

Lesion size was defined as the largest diameter of the lesion on diagnostic CT and CBCT images prior to puncture. Complications, such as pneumothorax or hemoptysis, were taken from the procedure’s record, from the patient’s medical records and from follow-up images. If treatment of a possible complication was necessary, such as the insertion of a thoracic drainage catheter in the event of pneumothorax, this was also recorded. The radiation dose was calculated as the total radiation exposure during the entire procedure. The measured dose/area in microGy/m^2^ and the total absorbed radiation dose (in milliGy) were recorded. In 27 cases, the radiation dose could not be registered due to the fact that these data were not collected in the machine. In our study, technical success was defined as the placement of the coaxial introducer tip at the target point in the CBCT images, or at a point close enough to it to allow access to the lesion with the axial needle. The final diagnosis of a target nodule was confirmed by pathological analysis of the samples. If the result showed malignant tumors or specific benign abnormalities, such as tuberculosis, these abnormalities were accepted as the final diagnosis. In the case of benign nonspecific abnormalities such as chronic inflammation, the lesions were considered negative and PTB was repeated or surgical resection was performed.

### 2.1. Ethics Committee

Taking into consideration that the work presented for publication is a part of a clinical audit related to quality, it was not necessary to request a favorable evaluation by an ethics committee, in accordance with the provisions of Section 5 of the Guide for Members of the Research. As indicated in the Guide, we would find ourselves needing a clinical audit to be carried out to “know if the best practices have been adopted”. Based on this, it is not strictly a research study subject to prior evaluation by a Research Ethics Committee, because it is not a research project, but part of a quality assurance study. Regarding the analysis and treatment of data conducted in this work, it is framed within one of the cases in which the General Data Protection Regulation (RGPD) allows the treatment of health data collected in its article. The aforementioned article states that “the treatment of health data will be allowed when the objective is to guarantee high levels of quality and safety of healthcare, especially in those cases in which the treatment is carried out by professionals subject to professional secrecy, as is the case here raised”.

All patients signed the informed consent form when performing the intervention.

### 2.2. Statistical Analysis

We used SPSS version 23.0 for Windows (SPSS Inc., Chicago, IL, USA). A descriptive analysis of the variables collected in the study was carried out. Quantitative variables were expressed as mean ± SD, median, and range. The qualitative variables were expressed as absolute and relative frequencies. The comparison of means between two groups was made using the Mann–Whitney U test after checking the assumption of normality using the Kolmogorov–Smirnov test. The association between qualitative variables was estimated by means of the Chi-square test or Fisher’s test. The risk factors associated with complications were determined using logistic regression models.

## 3. Results

In the present study, 98 PTBs were analyzed (Table 1), carried out on 94 patients with a mean age of 68.6 ± 9.05 years, with a predominance of males (73.4%). The mean nodule size in the 98 PTB performed in total was 35.6 ± 20.6 mm with a mean distance to the pleura of 1.3 ± 2.6 cm.

The mean radiation time was 4.9 ± 2.4 min.

### 3.1. Diagnostic Accuracy

Ninety-four biopsies were performed, of which 91 were technical successes (TS) and in three cases we were unable to position the coaxial needle at the desired point (technical failure: TF).

Of the 91 technical successes, in two cases the sample was doubtful or insufficient (FN) and the puncture had to be repeated. Another cause of repetition was a failure in the processing of the sample.

The three technical failures were diagnosed in the operating room, although a biopsy was previously repeated in one of them, resulting in a doubtful sample (TS, FN).

On the 98 total punctures performed, we obtained 87 diagnoses of malignancy, three of benignity, and an indeterminate diagnosis due to the total necrosis of the sample, also surgically diagnosed.

The overall sensitivity was 91.5% (87 diagnoses/95 TS = obtained samples) and our specificity was 100% (3/3).

There were a total of seven diagnostic failures: in three cases we were not able to position the needle at the desired point (TF), we had three cases with insufficient samples (FN), and an indeterminate diagnosis due to total necrosis of the sample. Four diagnoses were made in the operating room.

### 3.2. Complications Related to PTB and Its Influencing Factors

Of the total PTBs performed, 38 cases of pneumothorax (38.8%) were registered—97.4% (37 cases) of them mild and only one was clinically severe (Table 1). On five occasions, percutaneous treatment was performed with the placement of a drainage catheter and its aspiration (Figure 8).

Alveolar hemorrhage and emphysema in the puncture path were the two complications with the highest incidence: 17 (17.3%) and 4 (4.1%), respectively. We barely registered two (2%) cases of hemoptysis in the 98 punctures, but there were two (2%) cases of other types of bleeding, one of which required embolization of an intercostal artery. However, there were no cases of air embolism or mortality in our study.

The factors associated with the presentation of pneumothorax were analyzed (Table 2). No significant differences in age or sex were observed between punctures with and without pneumothorax; however, a higher percentage of complications was observed between males (86.8% vs. 18.4%; Odds Ratio (OR) = 2.1). The puncture position was significantly associated with pneumothorax, with a higher prevalence of complications being observed among punctures with a prone position, with a three-fold higher risk (OR = 2.8). More complications were detected in punctures performed in the fissure area (23.7% vs. 10.0%; *p* = 0.067), being at the limit of significance and showing a risk almost three times higher than in punctures not located in the area fissure (OR = 2.8; 95% CI = 0.9–8.6). On the other hand, punctures in the subpleural area showed a protective behavior against complications; those punctures located in said area presented 70% (OR = 0.3; 95% CI = 0.1–0.7) less risk of presenting pneumothorax compared to punctures in other areas.

The presence of pneumothorax was significantly associated with a smaller nodule size (OR = 0.9) and a greater distance from the nodule to the pleura (OR = 1.02).

Adjusting for the size of the nodule and the subpleural puncture area, it was observed that patients in the prone position had a 2.6 times greater risk of presenting pneumothorax-type complications than those in the supine position (OR = 2.61; 95% CI = 1.03–6.59). As in the univariate analysis, puncture in the subpleural area was observed as a protective factor, with a lower risk of pneumothorax than punctures in other locations (OR = 0.36).

### 3.3. Radiation

The means of the total estimated effective radiation dose of the patient and the product of the dose area were 567.5 milliGy (range: 94–1533 milliGy) and 11,722.4 microGy/m^2^ (range: 1534–3350 microGy/m^2^), respectively.

The mean time of scopia used was 4.99 min (range: 1.5–12.8 min).

## 4. Discussion

In our study, CBCT-guided PTB showed a diagnostic sensitivity of 91.5% (87 out of 95) and a specificity of 100% (3 out of 3). These results are comparable with the already published diagnostic accuracy of the CBCT-guided biopsies [12,16,18,19,20]. Lee et al. [18] performed 1153 PTB on 1108 patients and reported a sensitivity, specificity, and precision for the diagnosis of malignancy by CBCT of 95.7% (733 of 766), 100% (323 of 323) and 97% (1056 of 1089), respectively. We believe that the advantages of CBCT, such as flexibility in the selection of the entry site and the virtual trajectories overlapped on the real-time image of the fluoroscopy, contribute to a high diagnostic precision in small lesions. In addition, compared to the CT fluoroscopy system, the CBCT system allows synchronization of the patient’s breathing and the advancement of the needle, minimizing possible deviations, and allowing operators to avoid dangerous organs or ribs that could block a possible path puncture [6,16,20].

We had seven (7.1%) failed diagnoses in our study. One of them was considered a technical success but the pathologic analysis found only necrotic tissue and with another of the samples there was a processing failure. Three (3%) were technical failures because we were not able to place the coaxial puncture needle in a safe position to take the biopsies. In three (3%) punctures, pathologic analysis described normal lung or inflammatory changes that could suggest adjacent malignancy. Taking into account the mentioned diagnostic performance and technical error rate of PTB, we believe that CBCT-guided biopsies are highly profitable diagnostic methods in lung lesions and that their accuracy is comparable to that of CT fluoroscopic-guided biopsies.

We recorded a pneumothorax rate of 38.8% (38 of 98) and the incidence of catheter placement for thoracic drainage was 13.1% (5 of 38; 5.1% of all PTBs). These figures are within the previously published ranges for pneumothorax (12.0–44.6%) and the frequency of drainage catheters placement (0–32.7%) after a CT-guided biopsy, CT-guided fluoroscopy biopsy, or CBCT-guided biopsy [1,2,6,11,18,19,20,21,22,23,24,25,26]. The risk factors for pneumothorax were a greater distance to the pleura and the patient’s prone position. It seems logical to think that a longer puncture path is related to a greater possibility of suffering pneumothorax due to the rupture of a greater number of alveoli. It is also possible that, in a longer puncture, deviations of the needle occur, forcing it to be redirected [19]. Unfortunately, in our study we did not record the number of needle redirections. Our analysis, adjusting for the factors that are significantly associated with pneumothorax, found that the subpleural location is the only one with an independent impact to estimate risk, being a protective factor. Other authors [22] have postulated that a subpleural biopsy that forces the needle to be positioned in a transverse path supposes a greater risk of pneumothorax.

In our study, patients with a nodule placed in a lung segment next to a fissure have an almost three times higher risk of presenting a postpuncture pneumothorax than those with nodules in other locations, although this difference is not statistically significant (*p* = 0.074). As in the Shiekh or Cox studies [25,27], we did not detect differences between biopsies made in the upper or the lower lobes, which has been described by other groups [18,21]. The prone position of the patient was also associated with a higher frequency of pneumothorax. This could be due to the difficulty of the patients to maintain the necessary apneas during the advancement of the coaxial needle and the introduction and removal of the cutting needle. Other groups [18,20,25] have also reported a higher frequency of pneumothorax in patients in this position, although their studies have not detected significant differences. A larger size of the nodule seems to correspond to a lower possibility of suffering this complication. In the same sense, Cox et al. [27] described a strong correlation between the lesion size and the presence of postprocedure pneumothorax (*p* = 0.0044), appearing in 46.2% of the nodules smaller than 3 cm and in 22.2% of the bigger. Tunyarat et al. [19] published a study on 216 patients confirming these findings. Our work shows a higher male predominance in patients with pneumothorax compared to patients without pneumothorax (81.6% vs. 68.3%), according to other studies [18,21]. Probably due to an insufficient number of patients in our work, no statistically significant differences could be detected.

A hemoptysis rate of 2% (2 of 98) in our registry corresponds to that previously described (2.0–3.9%) on biopsies guided by CT or CBCT [2,16,20,22,24]. Although in our work it is difficult to find significant differences because of the small number of cases, we detected that a smaller size of the nodule was a risk factor (*p* = 0.030). We believe that biopsying small nodules can be associated with extraction of the adjacent healthy parenchyma causing local bleeding. This fact would also explain the greater frequency in our work of postpuncture alveolar hemorrhage in small nodules (*p* = 0.006), although in these cases bleeding has not been translated into hemoptysis. We also detected that the deepest location of the lesion was a risk factor for bleeding (*p* = 0.045), which can be explained in a similar way to that of pneumothorax because a long path of the needle has a greater chance of going through vascular structures. Several groups [18,19,20,22,24,28] have found similar findings in their observations. In our case, hemoptysis episodes were self-limiting and without clinical repercussions.

We believe that the CBCT virtual guide helps operators avoiding emphysema areas and visible vascular structures that can cause pneumothorax and hemoptysis according to previous studies [18,27]. It can also allow more accurate procedures without unnecessary redirection of the needles. In terms of radiation dose, the average dose/area per patient in our study was 11,722.4 microGy/m^2^ (range: 1534–33,503 microGy/m^2^). Unfortunately, estimated doses published in different works on CBCT are hardly comparable [20], although the superiority of this aspect of CBCT compared to biopsies performed with conventional CT seems to be proven [10]. In our hospital, we generally performed three CBCT per case, while other groups—for example, [12] perform Cone-Beam computed tomography scans only twice without postprocedure control. Nor have we collected anthropometric data such as patient weight or thickness. These aspects have a substantial influence on the dose received by the patients. The average fluoroscopy time in our work was 4.99 min (range: 1.5–12.80 min) and although these data were not collected, it seems logical to think that it is directly related to the number of needle repositions. In any case, the radiation dose in a PTB should not be a problem, although we must make continuous efforts to reduce the radiation dose through collimation. 

It seems that, in the future, radiology artificial intelligence (AI) will be a fundamental tool and imaging guide biopsy accuracy should be implemented in the coming years.

Machine learning (ML) is ready for the early detection of smaller and smaller lung nodules and the next step is to predict the behavior of precancerous lesions so the number of unnecessary biopsies will certainly be reduced [29].

Currently, there is a gap between advances in image acquisition hardware and image reconstruction software that can potentially be addressed by new deep learning (DL) methods to suppress artifacts, improve overall quality, and further reduce the radiation doses used. Furthermore, multimodal images in cancer and the association with quantitative functional measures, such as PET-MRI and PET-CT, have improved the precision of tumor detection and characterization [30] and we believe that in the not too distant future they will facilitate the puncture of “hot” nodules of increasingly smaller sizes and perfectly delimited from the surrounding tissues (atelectasis).

In pathology, DL algorithms have also gone a step further than pathologic diagnosis automation and have been used to characterize the underlying genotype–phenotype correlation within a tumor specimen [31] so the importance of diagnostic invasive techniques will remain basic.

There were several limitations to our study. First, it is a retrospective and nonrandomized design, therefore there may have been bias. We included patients studied consecutively in a single hospital and their amount was insufficient for detecting some statistically significant differences. The allocation of radiologists to the intervention room was also not randomized and the radiation dose received by them was not recorded. Additionally, as we mentioned previously, we had technical problems for registering the radiation dose in 27 patients. We believe that a prospective design with a greater number of patients is necessary to confirm our findings and a greater number of variables to assess radiation dose received by patients and operators.

## 5. Conclusions

The transthoracic biopsy performed with Cone-Beam CT is accurate and fairly safe when performed by experts for the diagnosis of lung lesions, alternative to conventional CT or CT with fluoroscopy, using a reasonable dose of radiation.

Complications are rare and the radiation dose used is not excessive. The most frequent complication was pneumothorax.

The main risk factor associated with the presence of pneumothorax, adjusting for the size of the nodule and the puncture area, was the prone position.

### Highlight


Cone-Beam CT-guided biopsy is a highly accurate and safe technique with a sensitivity of 91.5% and a specificity of 100%.Risk factors for pneumothorax are a deeper location of the nodule and prone position of the patient in the procedure.Alveolar hemorrhage and hemoptysis are the other usual complications and are more frequent in small and deeper lesions. 


## Figures and Tables

**Figure 1 diagnostics-10-01068-f001:**
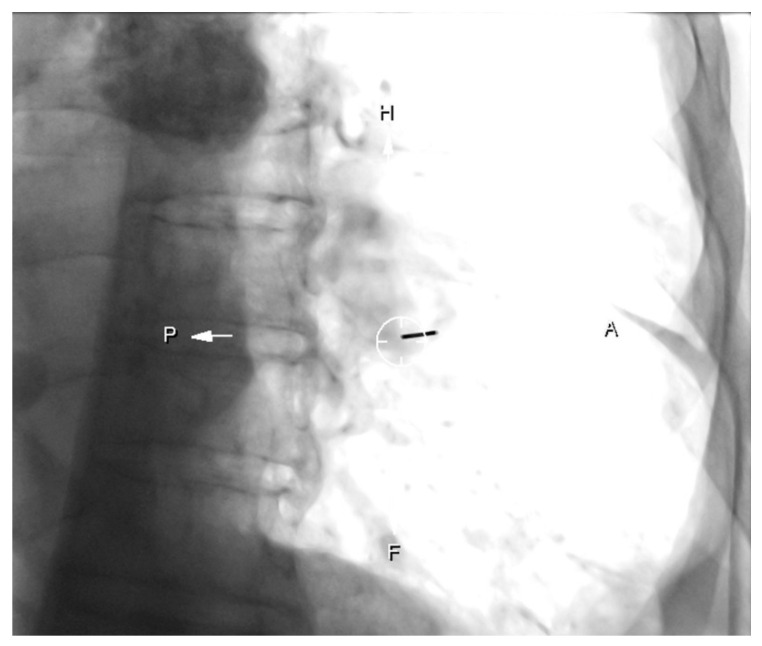
Patient with left low lobe mass. “Bulls-eye” projection: the coaxial needle is entering in the point of the skin indicated and aligned with the target point (circle).

**Figure 2 diagnostics-10-01068-f002:**
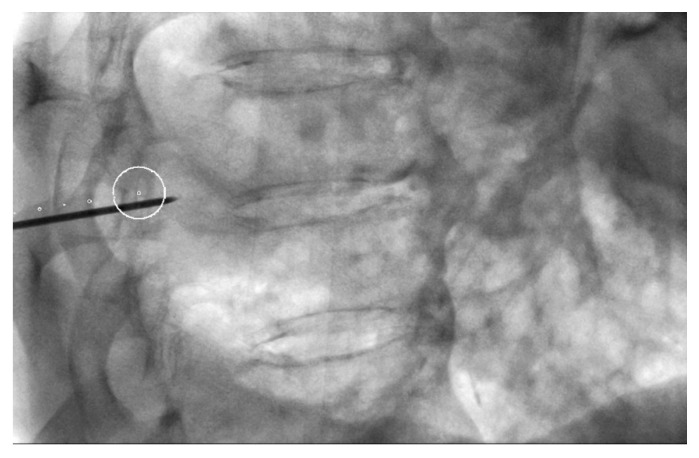
Lateral oblique projection on fluoroscopy. Virtual marks indicate the puncture route in caudal cranial direction and the circle the target. The circle is the point where the needle will reach.

**Figure 3 diagnostics-10-01068-f003:**
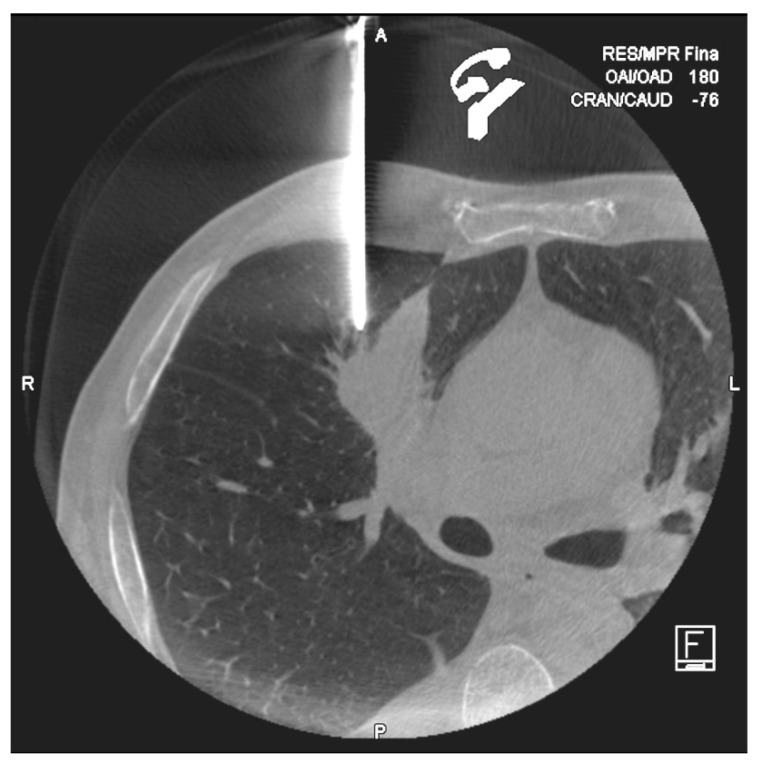
Cone-Beam computed tomography (CT) performed after coaxial introducer placement in a 58year old man. Control CT confirmed the exactly position of the needle tip.

**Figure 4 diagnostics-10-01068-f004:**
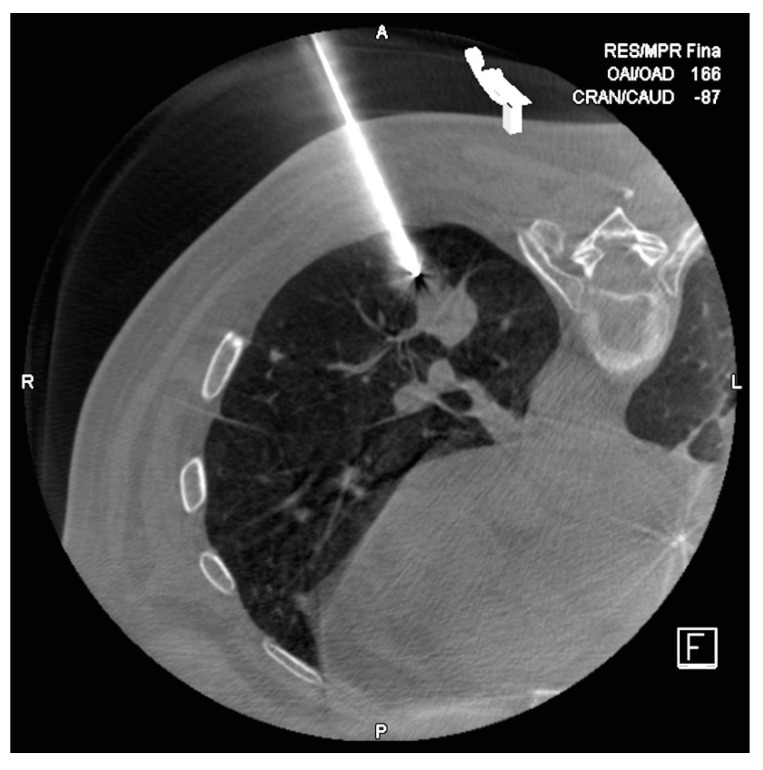
Lug nodule in lower left lobule of a 63 year old man. In the first control CT, we objectified the 17G needle perfectly positioned next to the target node.

**Figure 5 diagnostics-10-01068-f005:**
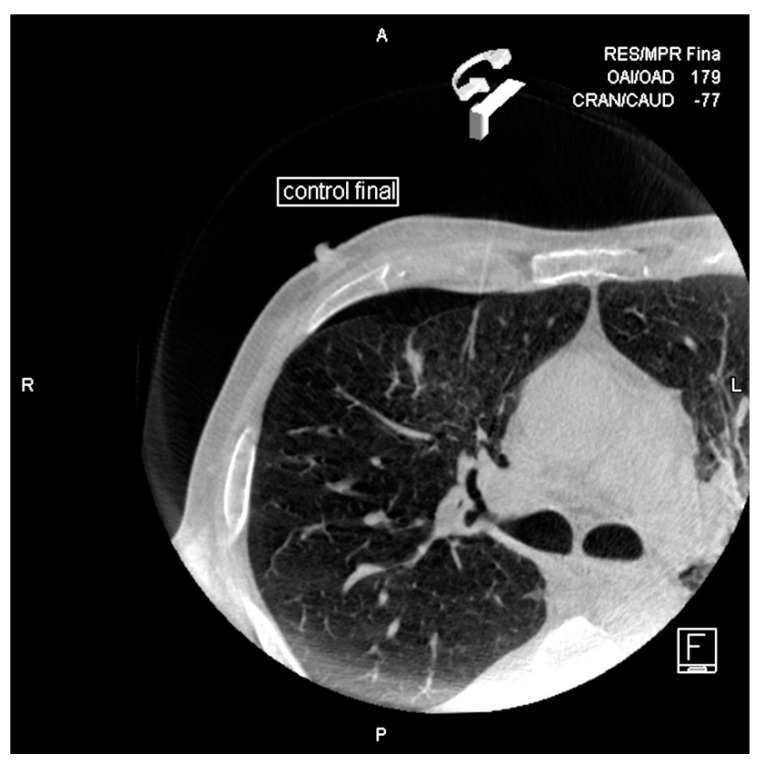
Small pneumothorax is observed after removing the needle.

**Figure 6 diagnostics-10-01068-f006:**
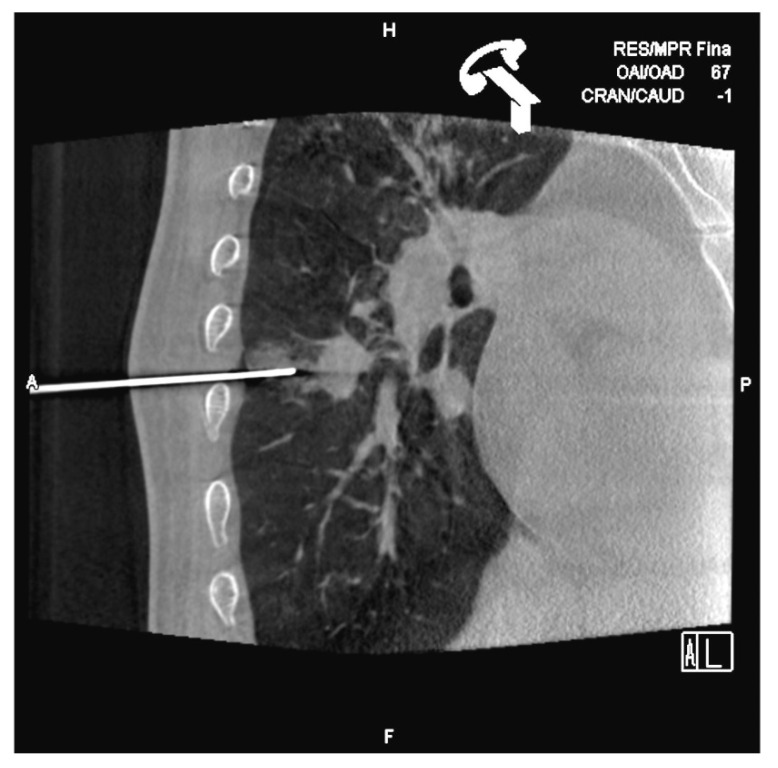
After inserting the coaxial needle, there is already a little hemorrhage observed in the punctured sane lung. We tried to pass the needle through the upper edge of the ribs to avoid the vascular nervous package.

**Figure 7 diagnostics-10-01068-f007:**
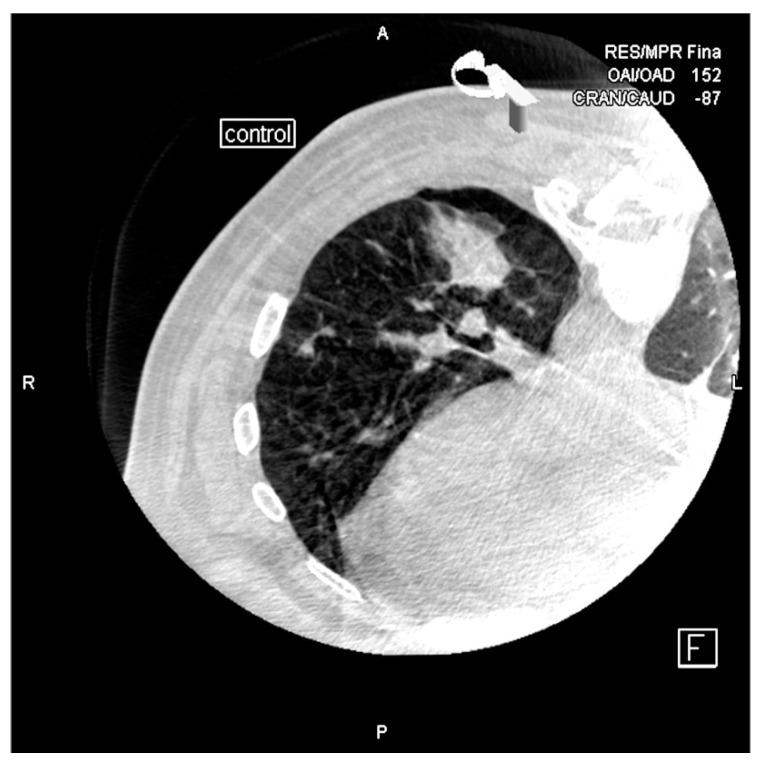
Alveolar hemorrhage and small pneumothorax are observed after removing the needle.

**Figure 8 diagnostics-10-01068-f008:**
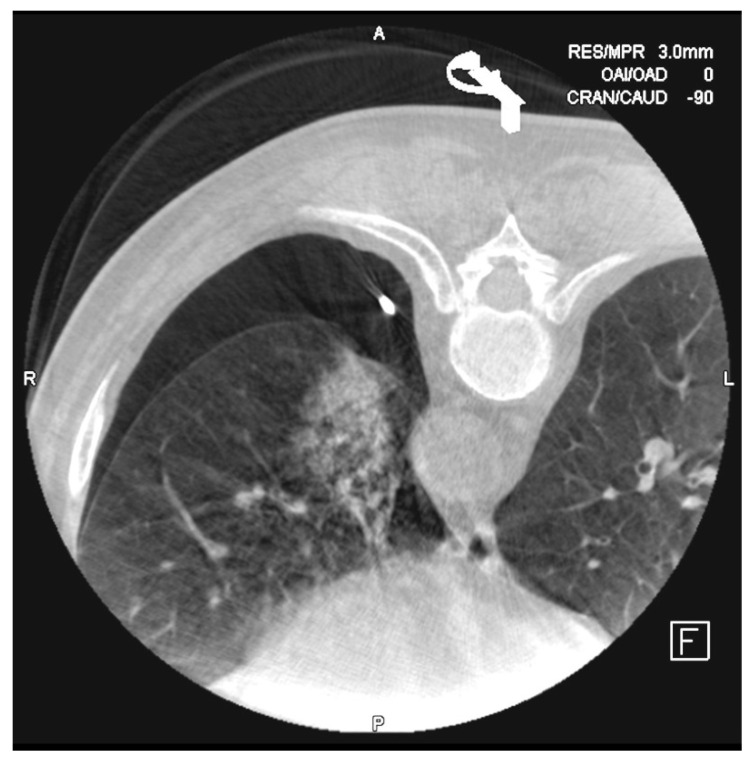
The most frequent complications after the biopsies are alveolar hemorrhage and pneumothorax. In this case, it was necessary to place a 6F drainage catheter for evacuating the air from the pleural chamber.

**Table 1 diagnostics-10-01068-t001:** Sample description.

Sample Description	*n* (%)	CI 95%	Mean ± SD	Median	Range
Gender					
Male	72 (73.5)	64.2–82.7			
Female	26 (26.5)	17.3–35.8			
Age	98		68.6 ± 9.03	69.6	42.4–89.2
Lesion size (mm)	98		35.6 ± 20.6	28.5	10–120
Distance from pleura to lesion (cm)	98		1.3 ± 2.6	0	0–15
Area dose (microGy/m^2^)	71		11,722.4 ± 6681.4	10,605	1534–33,503
Total dose (milliGy)	71		567.5 ± 329.9	538	94–1533
Radiation time (min)	71		4.9 ± 2.4	4.6	1.5–12.8

**Table 2 diagnostics-10-01068-t002:** Description of the sample according to the presence or absence of pneumothorax.

Sample	With Pneumothorax (*n* = 38)		Without Pneumothorax (*n* = 60)		*p*	OR (CI 95%)
	*n* (%)	Mean ± SD	*n* (%)	Mean ± SD		
Age	38	67.3 ± 8.1	60	69.5 ± 9.5	0.277	0.9 (0.9–1.0)
Gender						
Male	31 (81.6)		41 (68.3)			2.1 (0.76–5.5)
Female	7 (18.4)		19 (31.7)			1
Smoker					0.499	1.5 (0.5–4.6)
Yes	33 (86.8)		5 (13.2)			
No	49 (81.7)		11 (18.3)			
Patient position					0.020	
Supine	10 (26.3)		30 (50.0)			
Prone	28 (73.7)		30 (50.0)			2.8 (1.1–6.7)
Nodule location					0.476	
RUL	9 (23.7)		19 (31.7)			
ML	1 (2.6)		2 (3.3)			
RLL	6 (15.8)		8 (13.3)			
LUL	10 (26.3)		21 (35.0)			
LLL	12 (31.6)		10 (16.7)			
Paramediastinic	5 (13.2)		12 (20.0)		0.383	0.6 (0.2–1.9)
Cisure	9 (23.7)		6 (10.0)		0.067	2.8 (0.9–8.6)
Subpleural	15 (39.5)		41 (68.3)		0.005	0.3 (0.1–0.7)
Nodule size (mm)	38	30.0 ± 16.4	60	39.1 ± 22.4	0.030	0.9 (0.9–0.9)
Distance from pleura to nodule (cm)	38	1.4 ± 1.7	60	1.3 ± 2.9	0.007	1.02 (0.8–1.1)
Number of tissue samplings	38	2.1 ± 0.6	60	2.1 ± 0.4	0.955	
Hemoptysis						
Yes	0		38 (100)			
No	2 (3.3)		58 (96.7)			

LUL: left upper lobe; RUL: right upper lobe; LLL: left lower lobe; RLL: right lower lobe; ML: middle lobe; OR: Odds Ratio; 95% CI: 95% confidence interval.

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
