# Peer review of "Cone-Beam CT-Guided Lung Biopsies: Results in 94 Patients"

_diagnostics, 2020, doi:10.3390/diagnostics10121068_

Round 1

Reviewer 1 Report

This paper is an report on the use of CBCT to help a practitioner to guide a catheter for biopsy of the lung. Through a moderate radiation level, the guiding of the intervention allows a safer intervention for an expert practitioner.
The paper is interesting but does only provide "qualitative" results like "The transthoracic biopsy performed with ConeBeamCT is accurate and fairly safe performed by 269 expert hands for the diagnosis of lung lesions, alternative to conventional CT or CT with fluoroscopy, 270 using a reasonable dose of radiation".
The safety is linked to a moderate use of radiation and a reduction of pneumothorax and local hemorrhages.
The paper should be structured in a way that the compromises between radiation doses (compared to CT or fluoroscopy) and hemorrhages reduction should be more clearly presented (introduction and conclusion are not precise enough).
There should be also some prospective references about the use of AI to improve the process.

Author Response

Reviewer 1

This paper is an report on the use of CBCT to help a practitioner to guide a catheter for biopsy of the lung. Through a moderate radiation level, the guiding of the intervention allows a safer intervention for an expert practitioner.
The paper is interesting but does only provide "qualitative" results like "The transthoracic biopsy performed with ConeBeamCT is accurate and fairly safe performed by 269 expert hands for the diagnosis of lung lesions, alternative to conventional CT or CT with fluoroscopy, 270 using a reasonable dose of radiation".
The safety is linked to a moderate use of radiation and a reduction of pneumothorax and local hemorrhages.

The paper should be structured in a way that the compromises between radiation doses (compared to CT or fluoroscopy) and hemorrhages reduction should be more clearly presented (introduction and conclusion are not precise enough).

We have specified more the introduction and conclusions.

There should be also some prospective references about the use of AI to improve the process.

We have included a paragraph in the discussion part.

Reviewer 2 Report

Line 33 introduction-Thick needle percutaneous transthoracic biopsy (PTNB)- Change it to Just percutaneous transthoracic biopsy no need for descriptions like thick needle etc.

axial needle and automated biopsy system instead of biopsy forceps- this is not a forceps it is a needle.

“This system provides virtual marks on the fluoroscopic image to direct the puncture needle to the target marked on the source images”- Can you show a picture of this?

Your diagnostic accuracy paragraph is confusing, can you please re-write that with more clarity. And can you define diagnostic sensitivity and specificity please? Is it for malignant lesions? You define technical success as placement of the needle in the lesion which was 96.8%, but in the next line you say “3 patients had to have repeat biopsy due to insufficient sample” and not due to technical failure, it’s all very confusing. Also sensitivity 91.5% (87 out of 95) what is the 95 is it not 94? 7 cases of diagnostic failure: but you already had a benign malignant diagnosis in 90 patients (3 and 87 respectively)?

Author Response

Reviewer 2

  1. Line 33 introduction-Thick needle percutaneous transthoracic biopsy (PTNB) - Change it to Just percutaneous transthoracic biopsy no need for descriptions like thick needle etc.

As indicated, it has been changed. Appears in red in the document.

  1. Axial needle and automated biopsy system instead of biopsy forceps- this is not a forceps it is a needle.

It has been modified in the text.

  1. “This system provides virtual marks on the fluoroscopic image to direct the puncture needle to the target marked on the source images”- Can you show a picture of this?

The fluoroscopic images has been included in the methodology section.

  1. Your diagnostic accuracy paragraph is confusing, can you please re-write that with more clarity. And can you define diagnostic sensitivity and specificity please? Is it for malignant lesions? You define technical success as placement of the needle in the lesion which was 96.8%, but in the next line you say “3 patients had to have repeat biopsy due to insufficient sample” and not due to technical failure, it’s all very confusing. Also sensitivity 91.5% (87 out of 95) what is the 95 is it not 94? 7 cases of diagnostic failure: but you already had a benign malignant diagnosis in 90 patients (3 and 87 respectively)?

We have rewritten the paragraph again.